# Enhancing the Physical Activity Levels of Frail Older Adults with a Wearable Activity Tracker-Based Exercise Intervention: A Pilot Cluster Randomized Controlled Trial

**DOI:** 10.3390/ijerph181910344

**Published:** 2021-09-30

**Authors:** Justina Y. W. Liu, Rick Y. C. Kwan, Yue-Heng Yin, Paul H. Lee, Judy Yuen-man Siu, Xue Bai

**Affiliations:** 1School of Nursing, The Hong Kong Polytechnic University, Hong Kong 999077, China; rick.kwan@polyu.edu.hk (R.Y.C.K.); yueheng.yin@connect.polyu.hk (Y.-H.Y.); 2Research Institute for Smart Ageing, The Hong Kong Polytechnic University, Hong Kong 999077, China; xue.bai@polyu.edu.hk; 3Department of Health Sciences, University of Leicester, Leicester LE1 7RH, UK; paul.h.lee@leicester.ac.uk; 4Department of Applied Social Sciences, The Hong Kong Polytechnic University, Hong Kong 999077, China; judy.ym.siu@polyu.edu.hk

**Keywords:** frailty, physical activity, wearable activity tracker, cluster-RCT

## Abstract

A wearable activity tracker (WAT) incorporated with behavioral change techniques (BCTs) increases physical activity in younger adults; however, its effectiveness with frail older adults is unknown. The feasibility and preliminary effects of a WAT-based exercise intervention to increase physical activity levels in frail older adults was investigated in this pilot study involving 40 community-dwelling frail older adults. The experimental group received a 14-week WAT-based group exercise intervention and a 3-month follow-up, while the control group only received similar physical training and all BCTs. The recruitment rate was 93%, and the average attendance rate was 85.2% and 82.2% in the WAT and control groups, respectively, establishing feasibility. Adherence to wearing the WAT was 94.2% and 92% during the intervention and follow-up periods, respectively. A significant interaction effect between time and group was found in all physical assessments, possibly lasting for 3 months post-intervention. However, no significant difference between groups was observed in any daily activity level by the ActiGraph measurement. The majority of the WAT group’s ActiGraph measurements reverted to baseline levels at the 1-month follow-up. Thus, the WAT-based exercise program has potential for employment among community-dwelling frail older adults, but sustaining the effects after the intervention remains a major challenge.

## 1. Introduction

Frailty is conceptualized as a transitional state with multiple biologic systems sharing a common pathway that leads to a progressive decline in physiological reserve and performance [1]. Physical inactivity together with four other core clinical features, namely low grip strength, slow walking speed, self-perceived exhaustion, and unintentional weight loss, form the phonotype of frailty [2]. Frail older adults often demonstrate reduced strength and physiological malfunctioning, which increase an individual’s susceptibility to disability, dependency, institutionalization, and hospitalization [3].

There is increasing evidence to show that frail older adults who maintain a physically active lifestyle benefit in terms of improved physical endurance and functional status [4]. In one study a significant positive effect on frailty (i.e., reserve frailty status) was observed when sedentary time was replaced with moderate to vigorous physical activity (MVPA) but not with light-intensity physical activity [5]. To be considered physically active, it is recommended that older people engage in MVPA for at least 150 min per week, with each session lasting for at least 10 min or longer [6]. Despite the benefits of regularly performing physical activity, many older people still live a sedentary life, with a prevalence rate of over 60% in the United States [7] and over 40% in China [8]. The major obstacles that frail older adults face in adopting a physically active lifestyle include a lack of self-belief, low self-efficacy, and poor coping strategies [9].

Adopting behavioral change techniques (BCTs) in the design of an exercise training program is a common strategy to increase older people’s self-efficacy so that they remain physically active [10]. A meta-analysis showed that those exercise interventions that incorporated BCTs were associated with a greater increase in physical activity than those that did not [11]. BCTs, including goal setting, feedback and reward on performance, social support, and coaching were consistently associated with better outcomes in different studies aimed at increasing physical activity [12]. However, the traditional method of delivering BCTs through personal contact is costly, less flexible, and not sustainable due to limitations in time and venue [13]. In addition, the effect is likely to stop once the program has been terminated, and older people may resume their sedentary lifestyle [14].

Nowadays, wearable activity trackers (WAT) are becoming increasingly popular. Many studies have found that different types of WATs, such as Fitbit, Jawbone, and ActivePAL, have acceptable reliability and validity in recording daily steps and time spent on MVPA in a free-living environment [15]. WATs offer the potential to support lifestyle interventions to increase physical activity [12]. They can track daily activity levels and synchronize the data (such as step counts, time spent on MVPA, heart rate, etc.) to smartphones. They provide interactive BCTs that allow users to set exercise goals, self-monitor their progress, read their goal attainment, and seek social support [16,17,18]. These BCTs can be continuously delivered to users by WATs to increase the users’ motivation and exercise self-efficacy, thereby sustainably maintaining their physical activity levels with a minimum of human support [12,18,19]. Eventually, users could experience improvements in their physical endurance and functional status.

Previous studies found that some older adults were reluctant to use any WAT due to a fear of technology. This is because older adults did not grow up with computers or other digital devices [20]. To successfully encourage older people to use a WAT in their daily life to enhance their PA, additional supports may be needed. A systematic review showed that WAT interventions grounded in BCTs appeared to be more effective in enhancing physical activity and functional outcomes in older people immediately after the intervention when compared with the usual care or health information. However, the effects in the period (e.g., from 3 to 6 months) after the completion of the intervention remain inconclusive, especially when the supports given by the researchers have been withdrawn and older adults have been left to use the WAT on their own. In addition, its effects compared with those from a conventional exercise program remain unknown [21].

Therefore, the purpose of this study was to investigate the feasibility and acceptability of a WAT-based exercise program when used among frail older adults and to evaluate the preliminary effects of this intervention when compared with a conventional physical training intervention to increase physical activity levels among frail older adults. The specific objectives were:To determine the feasibility (i.e., recruitment rate, attrition rate, frequency of using the WAT, adherence to an exercise regimen) of the WAT-based exercise intervention to be employed in community-dwelling older people with frailty;To determine issues relating to the acceptability of the WAT-based exercise intervention.To explore the preliminary effects of the WAT-based exercise intervention on improving physical activity levels, physical endurance, and the motivation to engage in physical activity.

## 2. Methods

### 2.1. Trial Design

This study was a cluster randomized controlled trial (RCT) with two parallel groups. Clusters were district community health centers funded by and under the supervision of the Hong Kong Government, which met a specific set of standard regulations and criteria on environment and practices. Four district community health centers were recruited by a convenience sampling method and were randomized into either the WAT (i.e., the experimental) group or the control group at a 1:1 ratio. The methods employed in the trial followed the CONSORT Cluster Trials Checklist [22] and are reported in this section. This trial has been registered with ClinicalTrials.gov (NCT03538418).

### 2.2. Participants

Community-dwelling older people with frailty were the target population of this study. They were recruited through the abovementioned community centers from June 2018 to December 2019. The criteria for the inclusion of participants were: (1) community-dwelling older people aged 65 years or above; (2) able to communicate in Cantonese because they have to understand the instructions; (3) able to walk with or without an assistive device to complete the Timed Up and Go (TUG) Test because adequate mobility and balance ability are needed to complete the exercise training; (4) own a smartphone for daily use; and (5) in a pre-frail or frail state as determined by a Fried Frailty Index (FFI) score of 1–5, with a score of 1–2 indicating pre-frailty and a score of ≥3 indicating frailty [2]. We excluded participants who achieved adequate daily physical activity levels (e.g., through such activities as hiking and Tai Chi), as reflected by their having engaged in MVPA for ≥150 min per week in the previous four weeks [6].

### 2.3. Interventions

The intervention group received a group-based, 14-week WAT-based exercise intervention that consisted of the WAT-based training and physical training with the adoption of BCTs (for the implementation procedures please see Figure 1). The control group received only the physical training, which also involved the BCTs, plus a health talk to control the contact time between the interventionist and the control group participants.

#### 2.3.1. WAT-Based Components in the Intervention Group

A commercial tracker (i.e., Fitbit–Charge 2^®^) was provided to the participants in the intervention group for 6 months. Fitbit Charge 2^®^ (Fitbit Inc., San Francisco, CA, USA) is a wireless, wrist-worn, triaxial accelerometer, which has been proven to be a useful and reliable activity tracker in different studies with different population groups [23,24,25]. The activity levels of participants in the WAT group were assessed by their WAT (i.e., Fitbit Charge 2) over 6 months during the intervention and follow-up periods. Their activity levels were reflected by their average daily number of steps per week, average daily number of floors climbed per week, and average daily amount of time (in minutes) spent engaging in sedentary, lightly active, fairly active, and active activities over the week. A good level of criterion validity was shown by the correlation between the Fitbit steps with the visually counted steps (ICC = 0.88) [26] and the Fitbit heart rate with ECG in aggregate (r = 0.83) [27].

Participants were instructed to wear the tracker at all times, except during aquatic activities. The participants were also asked to walk around 7000 steps per day at the beginning of the program, gradually increasing to >10,000 steps towards the end of the intervention. There is sufficient evidence to prove that such an approach is beneficial to health [28].

Two hourly face-to-face sessions followed by weekly to monthly telephone sessions were arranged for the participants in the intervention group, the aim of which was to educate the participants on how to use different features of the tracker and how to synchronize the data with the associated mobile app to self-monitor their daily activity levels, view feedback, and set goals. Because studies have shown that older people may be fearful of new technologies and reluctant to use them [29,30], in the sessions arrangements were made to offer them support on dealing with technical issues and BCTs aimed at strengthening their motivation. The participants were taught to use the WAT to monitor their data every day, and to gradually increase their physical activity levels to eventually meet the recommended levels.

Many BCTs that are effective at increasing the self-efficacy and physical activity behavior of older people can be delivered by WATs [31]. These BCTs include “prompting a review of behavioral goals”, “providing rewards contingent upon successful behavior”, “alerting participants to self-monitor their behavior”, “encouraging a focus on past successes”, and “giving cues to action [32]. They were continually delivered to participants in the 14-week intervention phase. The additional support, which included face-to-face sessions and telephone follow-up sessions given alongside the WAT, was gradually withdrawn from the participants. Eventually, all additional measures to support the use of WATs were stopped. In the 3-month follow-up period, the participants used only the WATs to sustain the effects of the intervention on promoting physical activity.

#### 2.3.2. Physical Training in the Intervention Group

Twelve weeks of center-based structured physical training were provided to the WAT group. The training was provided once per week and each session lasted for 45–60 min. This physical training consisted of balance, resistance, and aerobic exercises with warm-up exercises at the beginning and cool-down exercises in the end. The design of this program was based on the recommendations of the American Heart Association [33] and has proven to be effective at improving the physical functions of frail older adults [34,35]. Exercise teaching videos and pamphlets were disseminated to encourage the participants to continually practice their exercises at home for at least 30 min for 5 days per week to meet the recommendation on physical activity [6]. Some BCTs, which are essential to increasing the self-efficiency and motivation of older people to engage in physical activities, but not those involving the WAT [12], were delivered to participants through human contact. They included action planning, problem-solving, feedback, and coaching from an interventionist [12].

#### 2.3.3. Physical Training in the Control Group

The only difference between the physical training provided to participants in the control group and that provided to the intervention group was that the former did not receive any BCTs delivered by the WAT, as they had not been given activity trackers. Other arrangements, i.e., physical training type and frequency, health talks schedule, and group size, were similar to those for the intervention group. The contents of the health talks included a discussion on the importance of regular physical exercise, strategies to increase daily physical activity levels, and ways of promoting gerontological health.

### 2.4. Outcomes

Effect outcomes for both the intervention and control groups were measured at T0 and T1. In order to explore any potentially longer effects from the WAT-based exercise intervention, only participants in the intervention group underwent a further assessment at 1 (T2) and 3 months (T3) after the completion of the intervention. This is a pilot study to test the preliminary effects of the intervention immediately after completion (T1). We hypothesized that the effects of the intervention would be sustained after the intervention; therefore, we set measurements at 1 month (T2) and 3 months (T3) to follow up on whether the effects were sustainable. The feasibility of the intervention was explored throughout the period of the intervention. During interviews at T1, the participants were asked questions about the acceptability of the intervention.

#### 2.4.1. Feasibility of the Program

The following measures were used to determine the feasibility of the program: the recruitment rate, the attrition rate, the WAT wearing adherence rate, the exercise adherence rate, and the incidence of adverse effects during physical training and practice. The recruitment rate referred to the number of participants who gave their consent to join the study over the number of eligible participants. In the pilot study, we defined success in recruitment as obtaining the consent of more than 80% of eligible older people to join the study. The attrition rate was indicated by the percentage of participants who withdrew from the study. We defined success in participant retention as having 80% of the participants in the study complete the final assessment. WAT wearing adherence was measured over the 6-month period of the study (i.e., the number of days that a participant wore the tracker and synchronized the data throughout the study period). We defined participants who met the 80% recommended wear time as having good adherence. Exercise adherence was assessed through the participants’ attendance in the training sessions. We defined good exercise adherence as attendance by the participants in >70% of all training sessions. Adverse effects referred to injuries, discomfort, or accidents that occurred during physical training and practice as reported by the participants.

#### 2.4.2. Acceptability of the Program

Information on the acceptability of the intervention to the participants (the WAT group) was collected using focus groups and a post-intervention questionnaire within two weeks after the completion of the WAT-based exercise program. Four focus groups, which included all of the participants in the WAT intervention, with 5–6 people per group, were conducted in the community health centers by a well-trained research assistant (RA). A semi-structured interview guide was used. For example, the participants were asked to comment on their experiences with the program, their perceptions of its positive/negative aspects, their concerns and difficulties with participating and adhering to the program, and ways of improving the intervention. A post-intervention questionnaire was designed to evaluate the quality of the program as perceived by the WAT group of participants. The questionnaire consisted of 25 items, which were designed by the research team. The participants were asked to rate their level of satisfaction with the WAT intervention, the technical support, and the BCT support. Each item was rated on a Likert scale ranging from 1 to 6 (i.e., 1 = “strongly disagree” to 6 = “strongly agree”).

#### 2.4.3. Effect Outcomes

The preliminary effects on the participants were assessed in terms of their (1) activity levels, (2) physical endurance, (3) frailty status, and (4) self-perceived confidence in engaging in exercise. The following describes the instruments and procedures that were used.

The activity levels of all participants were measured by an ActiGraph w-GT3X accelerometer that the participants were instructed to wear on their non-dominant wrist for 24 h per day for 7 days. The activity output of the ActiGraph is provided as counts per minute (cpm). Counts, also known as vector magnitudes, were exported in 1-min epochs. Non-wearing time was defined as 60 or more consecutive zero cpm, and a day was defined as valid if the wearing time was at least 10 h. Participants who provided data for at least four valid days were included in the analysis. Activity counts were categorized into specified levels of MVPA based on Kwan et al.’s cut-off points for older people of ≥4117.1 cpm, referring to moderate intensity levels or above [36].

The physical endurance of the participants was assessed using the 30-s Chair Stand Test [37], the Timed Up and Go (TUG) Test [38], and the Two-Minute Walk Test (2 MWT) [39]. The 30-s Chair Stand Test involves having participants go from sitting to standing as many times as possible within 30 s, with higher scores indicating more sitting/standing in a test interval and better lower limb muscle power. Depending on age and sex, older people can be at risk for falls if they are unable to perform a certain number of completions of sit to stand, ranging from less than 4 (for women aged 90–94 years) to 14 (for men in their 60 s) [40]. The TUG Test measures walking speed by taking the time (in seconds) spent by the participants to execute a series of walking tasks, including standing up from a chair, walking for three meters, turning around, walking back to the chair, and sitting down [38]. The longer the time taken to complete the test the greater the indication of poor functional mobility. Similar to the Six-Minute Walk Test (6 MWT) and the Twelve-Minute Walk Test (12 MWT), the 2 MWT measures the self-paced walking ability of participants, but the 2 MWT is particularly suitable for frail older adults who cannot manage the longer walking tests [39]. In various studies, the 2 MWT correlated highly with both the 6 MWT and 12 MWT, indicating that all three walking tests are similar in their ability to measure gait speed and exercise tolerance [39,41,42].

Frailty status was assessed by the FFI, which quantifies the phenotype of frailty according to five items, namely, an unintentional loss, a consistent feeling of exhaustion, a slow walk time, a reduced grip strength, and a low physical activity level. FFI scores range from 0–5. A score of 1–2 indicates pre-frailty and a score of 3 or above indicates frailty [2].

Exercise Self-efficacy refers to the participants’ self-confidence in their ability to exercise in a variety of circumstances. It was assessed using the nine-item Chinese Self-Efficacy for Exercise scale (CSEE) [43,44], with each item rated on a 10-point Likert scale from 0 = not confident to 10 = very confident. The total score ranges from 0–90 and is obtained by summing the responses to each item, with a higher score indicating higher self-efficacy for exercise. The CSEE was validated on 192 Chinese older people. Discriminant validity was shown by the CSEE total score, which significantly differentiated between individuals who did or did not regularly engage in exercise. The Cronbach’s α was 0.75, showing an acceptable level of internal consistency [43].

The motivation to engage in physical activity was assessed using the 19-item Chinese version of the Behavioral Regulation in Exercise Questionnaire-2 (C-BREQ-2). The C-BREQ-2 is a 19-item questionnaire that assesses five domains: motivation, external regulation, introjected regulation, identified regulation, and intrinsic regulation. Each item is rated on a 5-point Likert scale ranging from 0 = “not true for me” to 4 = “very true for me”. The five domain scores are obtained by summing up the items under each domain, with higher scores indicative of a better presentation of a particular domain. The C-BREQ-2 has shown good levels of internal consistency (Cronbach’s alpha = 0.71–0.83) [45].

### 2.5. Sample Size

The primary aim of this pilot study was to evaluate the feasibility and acceptability of the WAT-based exercise intervention for use among community-dwelling frail older adults. In this regard, a formal calculation of sample size might not have been required [46]. However, given that the secondary aim of this pilot study was to estimate the preliminary effects of the intervention, the sample size should be reasonable enough to provide a valuable estimation of parameters to calculate the sample size for a future main study [47]. Hertzog [48] suggested that a minimum of 20 participants per group would be sufficient for a pilot study to estimate a between-group effect size since it could yield confidence intervals, the lower limits of which could define the range of the power analysis. Hence, we decided that the sample size for this pilot study would be 20 for each group.

### 2.6. Randomization

To avoid contamination among individual participants from the same community care center, each center was viewed as one cluster and was randomized to either the WAT group or the control group in a 1:1 ratio by using simple randomization, based on computer-generated random numbers prepared by an independent statistician. Potential participants were identified by social workers at the center, based on eligibility criteria. A trained RA approached those individuals and assessed their eligibility to join the study. Participants from each center were placed in their center’s corresponding group to avoid contamination effects across participants. The group allocation was concealed from the researchers until sample recruitment and baseline measurements in each center were completed.

### 2.7. Blinding

The outcome assessor who was responsible for all outcome assessments in all measurement time points was blinded to the group allocation as well as to the purpose of the study.

### 2.8. Statistical Methods

The data were analyzed using the statistical package SPSS 27 for Windows. The feasibility markers described above were evaluated by a descriptive analysis. A comparison of the profiles, which included the demographic data and outcome assessments of the participants, was made between the groups at baseline using the Mann-Whitney U-test (for continuous variables) and a Chi-square test (for categorical variables). Age and gender have often been identified as associated with physical activity levels (i.e., MVPA) in older people [49]. Therefore, a generalized estimating equation (GEE), controlling for the clustering effect of elderly centers and adjusted for age and gender, was used to measure the changes in the outcome measurements within one week (T1) after the completion of the program to investigate the preliminary effects of the intervention over time. The fixed effects for the interaction (group by time) were used to study the effects of the intervention. GEE was also used to determine the within-group effect from T0 to T3 in the intervention group. Regarding the ActiGraph data, the analysis was controlled for the elderly centers and adjusted for age, sex, and number of valid wearing days. In the analysis of Fitbit data, it was controlled for the elderly centers and adjusted for age and percentage of wearing time. All outcomes were analyzed with an exchangeable covariance structure and found to have fitness. This implies that the covariances between observations on the same respondents are equal. The level of significance was set to 0.05.

Survey and focus groups were used to evaluate the acceptability of the intervention to the participants in the intervention group. A descriptive analysis was used to analyze the survey. All audio-taped interviews were transcribed into Cantonese by the RA and the transcriptions were checked for accuracy by two members of the research team before the analysis. A descriptive content analysis was used to analyze the interview data [50].

### 2.9. Ethical Issues

Ethical approval was obtained from the Human Subjects Ethics Sub-committee of The Hong Kong Polytechnic University (HSEARS20180320006). The written consent of the participants was obtained prior to the collecting of data.

## 3. Results

### 3.1. Participant Flow

Figure 2 shows a flowchart that depicts the participant screening and recruitment process in this study, which was based on the CONSORT statement. Forty older people were recruited from four community centers. Two centers were randomly allocated to the WAT group, and the other two to the control group. Among 77 older people who were assessed for eligibility, 34 (44.2%) did not meet the criteria for sample recruitment and 3 (3.8%) declined to participate due to a lack of interest in the program.

### 3.2. Baseline Data

As shown in Table 1, the participants in both groups were predominately female. There were no significant differences between the groups in the majority of their demographic characteristics with the exception of age (72.1 ± 3.7 vs. 80.4 ± 6.83 years old, *p* < 0.001), marital status (*p* < 0.001), education (*p* < 0.001), hospitalization in the past 12 months (*p* = 0.046), and the use of walking aids (*p* = 0.002). In general, the participants in the WAT group were significantly younger and had received a higher level of education than those in the control group. More participants in the WAT group had stayed married, did not use any walking aids, and had no history of hospitalization in the past 12 months (Table 1).

The participants in the WAT group showed significantly better physical endurance, as shown by their results in the TUG Test (9.8 s vs. 15.6 s, *p* = 0.002), the 30-s Chair Stand Test (11.4 s vs. 7.5 s, *p* = 0.023), and the Two-Minute Walk Test (99.8 m vs. 77.1 m, *p* = 0.011) at baseline. They also exhibited significantly less severe characteristics of frailty, as reflected by their FFI scores (1.5 vs. 2.4, *p* < 0.001). There were no significant differences between the two groups in other measurements, including those measured by the ActiGraph, except in the average number of minutes taken to perform MVPA per valid day over the week (218.9 min/day/week vs. 145.6 min/day/week, *p* = 0.039). The WAT group spent significantly more time on MVPA than did the control group (Table 2).

### 3.3. Main Results

#### 3.3.1. Objective #1: Feasibility of the WAT-Based Exercise Intervention

The recruitment rate was 93% (i.e., 40 out of 43 eligible older people were recruited from the four community centers). The WAT group’s rates of adherence to wearing the activity tracker during the 14-week intervention and the 3-month follow-up period were 94.2% (92.3 out of 98 days) and 92% (i.e., 77.3 out of 84 days), respectively. Table 3 shows the change in the data collected by the WAT from baseline to 24 weeks. A statistically significant time effect was only found in the average number of stairs climbed daily (*p* = 0.039), with an average increase of 0.027 from baseline to 24 weeks. No statistically significant differences among the participants in the WAT group were found in the other data collected by the WAT. No participant withdrew from either the WAT or the control group. The attrition rate was 0%. The mean attendance in the WAT group and the control group was 11.9 ± 3.0 (85.2%) and 11.5 ± 3.2 (82.2%) out of 14 face-to-face sessions, respectively. There was no significant difference in attendance between the two groups (*p* = 0.551). No adverse event was report in either group.

#### 3.3.2. Objective #2: Acceptability of the WAT-Based Exercise Intervention

In the questionnaire shown in Table 4, and in the focus group in which all participants in the WAT group gave their feedback (*n* = 22), 20 participants (90.9%) reported that they used their tracker to monitor their activity levels every day. Only two participants (9.1%) reported using the tracker about 4 to 5 days per week. Twenty participants (all except for two) agreed or strongly agreed that the WAT was easy to use. However, the associated mobile app was less frequently used, with only 13 participants (59.1%) reporting that they used it every day. Seventeen participants (80.9%) agreed or strongly agreed that the app was clear and easy to use. Three participants also mentioned in the focus group that they used the apps were less frequently because the WAT already gave them sufficient information on their daily exercise performance. One participant even mentioned that he did not know how to use most of the features in the app. More than 85% of the participants agreed or strongly agreed that different functions in the WAT and the app could enhance their exercise self-efficacy, leading them to exercise regularly and, particularly, helping them in the areas of formulating personal exercise goals (Q7), prompting a review of pre-set goals (Q10), self-monitoring their exercise performance (Q16 and Q17), receiving biofeedback (Q18), and providing rewards contingent upon achieving the goals (Q24). Many participants also mentioned in the focus group that the daily self-monitoring of step counts could encourage them to do more exercise, as they wanted to achieve their exercise goals. Two participants mentioned that they would deliberately walk more in the evenings to meet their targeted daily exercise goals. Another participant mentioned in the focus group that he felt a sense of accomplishment in increasing his daily step goal from 8000 to 15,000 toward the end of the program. Two participants mentioned in the focus group that they were encouraged by the badges that they collected upon achieving their exercise goals. In addition, 19 (86.4%) agreed or strongly agreed that they would continue to use the tracker. At the same time, the support, guidance, and feedback demonstration given by the instructors either during the face-to-face sessions or in the telephone follow-ups (Q12, 14, 21) were highly valued by more than 85% of the participants. Most participants agreed or strongly agreed (89.6%) that they would recommend the program to other older people (Q25).

#### 3.3.3. Objective #3: Preliminary Effects of the WAT-Based Exercise Intervention

Table 5 present the comparisons that were made between the intervention and the control groups. A significant interaction effect between time and group was found in all of the physical assessments, which included the TUG Gest (Intervention vs. Control: 9.76 to 15.62 s vs. 7.98 to 14.55 s, *p* = 0.012), the 30-s Chair Stand Test (Intervention vs. Control: 11.36 to 7.50 s vs. 14.14 to 8.5 s, *p* = 0.002), and the Two-Minute Walk Test (Intervention vs. Control: 99.84 to 77.11 m vs. 116.23 to 80.57 m, *p* = 0.001) in T0 and T1. Significantly better performances in these physical assessments were found between the baseline (T0) measurement and the immediate post-intervention measurement (T1) in the WAT group, but not in the control group. A significant interaction effect between time and group was also found in some self-reported assessments, which included the FFI (Intervention vs. Control: 1.55 to 2.44 vs. 1.36 to 2.36, *p* = 0.038), the Amotivation component of BREQ-2 (Intervention vs. Control: 0.92 to 1.56 vs. 1.82 to 1.94, *p* < 0.001), and the identified regulation component of BREQ-2 (Intervention vs. Control: 11.29 to 12.30 vs. 11.39 to 10.00, *p* = 0.008). However, a significantly better performance was only found in the Amotivation domain of BREQ-2 between the baseline (T0) and the immediate post-intervention measurement (T1) in the WAT, but not in the control group. A significantly poor performance was found in the Identified Regulation domain of BREQ-2 between the baseline (T0) and the immediate post-intervention measurement (T1) in the control group (Table 5). No significant difference between the groups over time in any of the measurements made by the ActiGraph was observed in T0 and T1 (Table 5). However, a significant decrease in time spent in MVPA and peak cadence per week between T0 and T1 was identified only in the control group.

Table 6 shows the time effect of the WAT group on all outcome measurements over T0 to T3. Statistically significant time effects were found in all of the assessments on physical endurance, which included the TUG Test (T0:T1:T2:T3 = 9.76:7.98:7.62:8.23 s, *p* < 0.001), the 30-s Chair-Stand Test (T0:T1:T2:T3 = 11.36:14.14:13.40:15.22 s, *p* < 0.001), and the Two-Minute Walk Test (T0:T1:T2:T3 = 99.84:116.23:115.73:112.00 m, *p* < 0.001). Statistically significant changes in frailty status (T0:T1:T2:T3 = 1.55:1.36:1.20:1.22, *p* = 0.001) were found from T0 to T4. Statistically significant time effects were found in the majority of domains in the BREQ, with the exception of the Intrinsic Regulation domain. In addition, no statistically significant changes were found from T0 to T4 in the CSEE in the intervention group. With regard to the ActiGraph measurements, statistically significant time effects were found in the maximum duration of brisk walking per week (T0:T1:T2:T3 = 9.23:9.00:10.89:10.89 min, *p* = 0.003) and peak cadence per week (T0:T1:T2:T3 = 116.27:112.05:108.26:114.11 steps/min, *p* = 0.048).

## 4. Discussion

The major findings in this study are that the WAT-based exercise intervention has the potential to be implemented among older people with frailty. The recruitment rate of 93% in this study was higher than the recruitment rates of 24.7% to 66.7% that were reported in other similar pilot studies [51,52]. All participants completed the final assessments and no one dropped out, in contrast to similar studies, which had attrition rates of between 8% and 20% [51,52]. The participants in the study had a high level of engagement, with a good attendance rate in all of the face-to-face sessions. The rate of adherence to wearing the tracker was 94.2% during the intervention period. This rate was slightly lower, at 92%, during the 3-month follow-up period in the intervention group. We believe that once the participants in the intervention group became used to monitoring their individual daily activity levels using the trackers, the majority of them maintained this habit even after the conclusion of the program. Also, most of the participants were satisfied with their experience of using the WAT and agreed to keep using it. Most stated that they would recommend the program to other frail older adults. Finally, no adverse incidents occurred during the program. These findings provide preliminary evidence that this WAT-based exercise program is acceptable and feasible for older people with frailty.

This study also found a significant interaction effect between time and group in all of the physical assessments conducted immediately after the intervention period (comparing T0 and T1). The post hoc tests showed a significant within-group improvement in all physical assessments in the WAT group but not in the control group. However, we also observed that there were significant differences between the groups in all of these physical assessments at baseline. This should be considered when interpreting the results. When exploring the longer intervention effects in the WAT group, a significant time effect in all physical assessments was identified. An upward trend was observed in all of these assessments between immediately after, to 1 month and 3 months after the WAT-based exercise program.

In this study, no significant interaction effect between time and group was found in any of the ActiGraph parameters that measured the participants’ daily activity levels when comparing the measurements obtained between T0 and T1. Statistically significant differences between the groups were not interpretable due to the underpowered nature of a pilot study. Effect sizes (0.34 for step counts, 0.12 for the average duration of brisk walking, and 0.37 for MVPA) suggest that a larger-scale implementation of the intervention will likely produce significant improvements in the daily physical activity levels of frail older adults. Given that our WAT participants were frail, we considered it already very good that they had increased the number of steps that they took daily by approximately 890 and spent eight more minutes on MVPA by the end of the program, while the control group maintained a similar level of activity as before the program or declined in performance. This finding is similar to that of another pilot study that aimed to explore the preliminary effects of a wearable technology physical activity intervention for middle-aged and older adults. The participants in that study increased their activity by about 1090 steps, with an effect size of 0.35, by the end of the program [52]. In fact, a trend of increase was observed in four out of six ActiGraph parameters, namely in “average minutes spent on MVPA per valid day per week”, “average minutes spent on MVPA with an interval of >10 min”, “average duration of brisk walking per week”, and “average step count per week”. By contrast, the post hoc tests showed a significant reduction in average minutes spent on MVPA per valid day per week and in the duration of peak cadence per week in the control group.

To determine whether the effects could be extended to the 1-month and 3-month follow-up periods, a significant time effect was identified in two ActiGraph measurements, namely “maximum duration of brisk walking over the week” and “the duration of the peak cadence per week”. When scrutinizing the data, an upward trend was observed only in “maximum duration of brisk walking over the week”, while a downward trend was identified in “the duration of peak cadence per week”. In addition, at the 1-month follow-up period the majority of the ActiGraph measurements of the participants in the WAT group dropped back to the baseline levels. Similarly, over the 24-week study period, with the exception of the average number of stairs climbed per week as measured by the participants’ own WAT (i.e., Fitbit charge 2), there was no change in the other parameters, although the majority of the participants continued to use their WAT.

These findings demonstrate that the participants were adhering to the recommendations and had gradually increased their activity levels during the 14-week intervention phase. Our results showed that the WAT-based exercise intervention was more effective at increasing physical activity levels than was the case when no WAT was provided, as seen by the performance of the control group, but less success was seen in the subsequent 3-month follow-up phase. This finding was similar to that in Bickmore’s study [53], where the participants failed to maintain the habit of exercising regularly during the 10-month follow-up period. The experiences of the participants in Bickmore’s study were assessed after two main phases of the RCT: an intervention phase and a follow-up phase. During the 10-week intervention phase, the participants had regular contact with an RA to provide structured technical support for using their WAT. During the 6-month follow-up phase, the participants were left on their own to use their own WAT. All participants’ ratings of their experience of using the WAT, which included “ease-of-use”, “usefulness”, and “acceptance”, showed a downward trend between the intervention and follow-up phases. These findings indicate that older people may need more support to continue to use the WAT to reach the point of achieving increased MVPA [54].

Bickmore’s [53] study also showed that those participants with high levels of digital literacy tended to continue their exercise habit with encouragement from the exercise coaching online platform when compared with those participants with low digital literacy. Thus, a possible reason for why the participants’ activity levels were not maintained in the follow-up phase of our study was the low digital literacy of our participants. Once technical support was withdrawn by the research team, technical issues that arose were not resolved in a timely manner. The participants might only have been able to use the most basic functions of the WAT, which affected the delivery of different kinds of BCT to sustain their physical activity levels. Other possible reasons include the possibility that the changes in behavior exhibited in the intervention phase were mainly caused by their interaction with the physical instructors. The findings in the survey suggest that our participants cherished the guidance, feedback, demonstration, and support given by the physical instructors. Thus, after the withdrawal of instructions offered through human contact, the participants lost their motivation to keep up their increased levels of physical activity and relapsed to earlier levels.

Some preliminary effects on the participants’ behavioral regulation with regard to regular engagement in exercise were also observed from the WAT-based exercise program. We saw an upward trend in all five domains under BREQ-2. A significant time effect between the intervention and follow-up phases (from T0 to T3) was also identified within the WAT group in four out of the five BRECQ-2 domains. This means that in most areas behavioral regulation improved when compared to the baseline. Some of the improvements seem to have been maintained until 1 and 3 months after the program. However, a significant interaction effect between time and group was only identified in the domains of “Amotivation” and “Identified regulation”. Identification involves a conscious acceptance of exercise as being important to achieving positive health outcomes. In this domain, the measurements of the WAT participants remained similar between baseline and post-intervention, whereas the scores obtained by the control group fell greatly between baseline and post-intervention. Amotivation refers to a failure to value exercise. We hypothesized that after both groups had received the interventions, the participants would come to value the beneficial effects of exercise more than they had prior to their participation in the program. Unexpectedly, the scores in this domain increased in both groups, but more so in the control group. In a future study, it may worth exploring what the key factors are that affect older people’s behavioral regulation with regard to regular exercise, such as which types of BCTs might be effective.

There is ample epidemiological to suggest that even small increases in physical activity among older people can produce a large improvement in health [55]. Although our objective measurements obtained by using the ActiGraph showed that the participants did not maintain their levels of physical activity during the follow-up period, the significant improvement in their physical endurance as reflected by the results of the physical assessments lasted for 3 months after the program when compared with the baseline. In the future, we can explore ways to incorporate more strategies to sustain the activity levels of frail older adults even after the end of an intervention, to extend the beneficial effects of exercise on their physical health. Those strategies may include involving young family members who are familiar with technology. This may be particularly important for older people with low digital literacy. In addition, ease of use and offering a variety of devices to older people with various levels of digital literacy should be considered when selecting technologies (i.e., WAT). Given that our participants cherished the interactions that they had with the physical instructors, we might also consider offering some booster group sessions during the follow-up period. Social support is associated with the maintenance of physical activity among older adults [56]. Social interactions, either with other participants or with family or friends or a physical instructor, could be a powerful tool for increasing the efficacy of a WAT.

This study has a number of limitations that warrant acknowledge and consideration. First, the sample size was small, which may lead to insufficient statistical power to evaluate the true effects of the intervention. However, the major purposes of this pilot study were to explore the feasibility, acceptability, and possible preliminary effects of the intervention instead of the effects of the intervention. Second, we observed significant differences between the experimental and control groups at baseline, particularly in their physical assessments (i.e., age, sex, and walking aids). Therefore, the significant group X time effects identified after the completion of the program must be interpreted with caution. As mentioned, the aim of this pilot study was to explore the possible preliminary effects of the intervention. Therefore, the true effects of the intervention should be evaluated in a full-scale clinical trial with a larger sample size. Third, there was no passive control group to comprehensively compare the effects of the WATs + BCTs with usual care, which may be addressed in a future study.

## 5. Conclusions

In conclusion, this 14-week WAT-based exercise program has the potential to be employed among community-dwelling frail older adults with various levels of digital literacy. Sustaining the effects of the intervention on the daily activity levels of frail older adults after the completion of the supervision sessions remains a major challenge in this kind of research study. The provision of more support from family members and physical instructors, especially during the follow-up phase, may help to sustain adherence to the recommended levels of daily activity.

## Figures and Tables

**Figure 1 ijerph-18-10344-f001:**
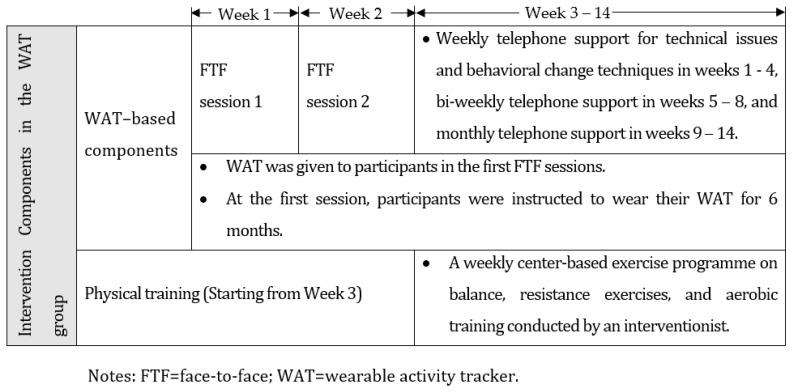
Intervention components for the WAT group.

**Figure 2 ijerph-18-10344-f002:**
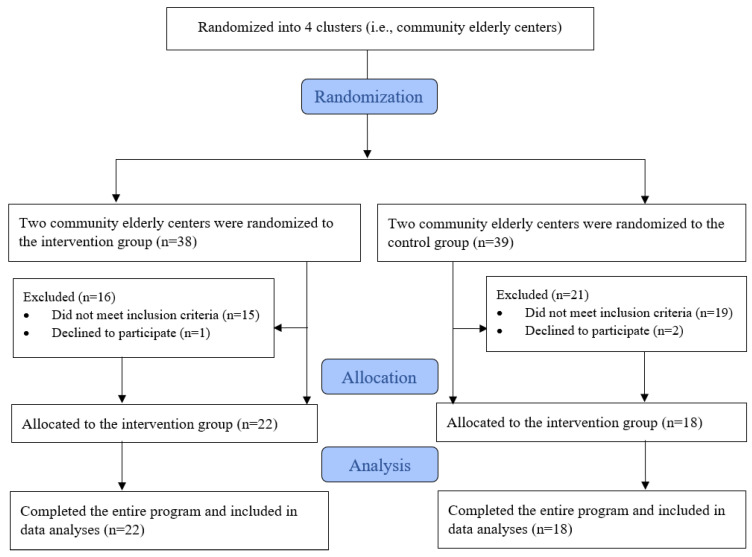
CONSORT diagram through the phases of the study.

**Table 1 ijerph-18-10344-t001:** Demographic characteristics of the participants.

		Total		Intervention (*n* = 22)		Control (*n* = 18)		*p*
		*n*	(%)	*n*	(%)	*n*	(%)	
Age [mean ± SD, median]		[75.9 ± 6.74, 73.5]		[72.1 ± 3.7, 71.5]		[80.4 ± 6.83, 80]		<0.001
Gender								0.130
	Male	6	(15.0)	5	(22.7)	1	(5.6)	
	Female	34	(85.0)	17	(77.3)	17	(94.4)	
Marital status								0.001
	Not married	5	(12.5)	5	(22.7)	0	(0.0)	
	Married	17	(42.5)	12	(54.5)	5	(27.8)	
	Widowed	16	(40.0)	3	(13.6)	13	(72.2)	
	Divorced/Separated	2	(5.0)	2	(9.1)	0	(0.0)	
Education								<0.001
	No formal education	8	(20.0)	1	(4.5)	7	(38.9)	
	Primary level	11	(27.5)	2	(9.1)	9	(50.0)	
	Secondary level	16	(40.0)	14	(63.6)	2	(11.1)	
	Degree level	3	(7.5)	3	(13.6)	0	(0.0)	
	Others	2	(5.0)	2	(9.1)	0	(0.0)	
Housemate								0.575
	Live alone	16	(40.0)	9	(40.9)	7	(38.9)	
	Live with family	19	(47.5)	11	(50.0)	8	(44.4)	
	Live with others	5	(12.5)	2	(9.1)	3	(16.7)	
Hospitalization in past 12 months								0.046
	No	37	(92.5)	22	(100.0)	15	(83.3)	
	Yes	3	(7.5)	0	(0.0)	3	(16.7)	
Number of medications taken in past 12 months								0.056
	0	3	(7.7)	3	(14.3)	0	(0.0)	
	1	10	(25.6)	6	(28.6)	4	(22.2)	
	2	6	(15.4)	4	(19.0)	2	(11.1)	
	3	11	(28.2)	7	(33.3)	4	(22.2)	
	4	6	(15.4)	0	(0.0)	6	(33.3)	
	5 and more	3	(7.7)	1	(4.8)	2	(11.1)	
Emergency service in past 12 months								0.271
	No	30	(75.0)	18	(81.8)	12	(66.7)	
	Yes	10	(25.0)	4	(18.2)	6	(33.3)	
Medical follow-up in past 12 months								0.810
	No	34	(85.0)	3	(13.6)	2	(11.1)	
	Yes	6	(15.0)	19	(86.4)	16	(88.9)	
Private hospitals follow-up in past 12 months								0.789
	No	34	(85.0)	19	(86.4)	15	(83.3)	
	Yes	6	(15.0)	3	(13.6)	3	(16.7)	
Walking aids								0.002
	None	32	(80.0)	22	(100.0)	10	(55.6)	
	Walking stick	2	(5.0)	0	(0.0)	2	(11.1)	
	Crutches	6	(15.0)	0	(0.0)	6	(33.3)	

**Table 2 ijerph-18-10344-t002:** Outcome measures at baseline.

Outcome Measure [Mean ± SD, Median]	Total	Intervention (*n* = 22)	Control (*n* = 18)	*p*
TUG (second)	12.4 ± 6.27, 10.39	9.8 ± 2.47, 8.98	15.6 ± 7.92, 13.2	0.002
30-s Chair Stand Test (second)	9.6 ± 4.79, 10	11.4 ± 3.99, 11	7.5 ± 4.93, 8.5	0.023
Two-Minute Walk Test (meter)	89.6 ± 26.47, 86.25	99.8 ± 22.82, 91.5	77.1 ± 25.75, 78	0.011
FFI	2 ± 0.81, 2	1.5 ± 0.67, 1	2.4 ± 0.7, 2	<0.001
CSEE	37.2 ± 17.08, 39.5	43.6 ± 15.73, 45.5	29.4 ± 15.72, 25.5	0.014
C-BREQ-2 Amotivation (Sum of Q5, 9, 12, 19)	1.21 ± 2.4, 0	0.92 ± 1.86, 0	1.56 ± 2.94, 0	0.854
C-BREQ-2 External Regulation (Sum of Q1, 6, 11, 16)	2.52 ± 2.98, 2	2.73 ± 2.96, 2	2.28 ± 3.06, 1.5	0.641
C-BREQ-2 Introjected Regulation (Sum of Q2, 7, 13)	3.12 ± 3.04, 2.5	3.78 ± 2.89, 3.5	2.33 ± 3.10, 0	0.138
C-BREQ-2 Identified Regulation (Sum of Q3, 8, 14, 17)	11.74 ± 2.43, 12	11.29 ± 2.73, 11.5	12.30 ± 1.92, 12	0.195
C-BREQ-2 Intrinsic Regulation (Sum of Q4, 10, 15, 18)	10.8 ± 7.24, 12	10.82 ± 4.56, 11	10.78 ± 3.95, 12	0.977
*ActiGraph*				
MVPA (mins per valid day per week)	185.9 ± 103.73, 166.48	218.9 ± 112.24, 216.5	145.6 ± 77.44, 142.64	0.039
MVPA (mins per valid day per week, only include >10 min)	37.5 ± 38.14, 22.85	47.8 ± 44.68, 34.64	25 ± 23.85, 22.1	0.157
Maximum duration of brisk walking ^#^ over the week (mins)	7.7 ± 7.92, 5	9.2 ± 8.8, 7.5	5.8 ± 6.42, 3.5	0.189
Average duration spent in brisk walking ^#^ per week (mins)	2.7 ± 3.46, 1.5	3.3 ± 3.88, 1.93	1.8 ± 2.77, 0.71	0.140
Peak cadence (steps/min) per week	113.8 ± 18.19, 116	116.3 ± 19.4, 118	110.8 ± 16.64, 111.5	0.301
Average step count per week (steps)	11,691.8 ± 3965.77, 11,335.87	12,787.3 ± 4261.92, 12,139.29	10,352.9 ± 3192.38, 10,266.29	0.073
Average duration of true sleep per week (mins)	402.9 ± 77.4, 402.38	385.7 ± 79.39, 396.6	423.8 ± 71.51, 417.79	0.174
Average duration of total sleep per week (mins)	482 ± 80.79, 478.54	462.3 ± 81.17, 451.57	506.2 ± 75.59, 499.79	0.100
Average number of days with 7 h of sleep per week	2.9 ± 2.03, 3	2.6 ± 2.06, 2	3.3 ± 1.99, 3	0.310

TUG, Timed Up and Go Test; FFI, Fried frailty index; CSEE, Chinese self-efficacy for exercise; C-BREQ-2, Chinese Behavioral Regulation in Exercise Questionnaire-2; MVPA, Moderate to vigorous physical activity. ^#^ Brisk walking was defined as >100 step/min.

**Table 3 ijerph-18-10344-t003:** Baseline to 24-week changes in the data collected by the WAT ^a^.

	B	SE	95% CI	*p*
Count of average daily steps over the week (steps)	38.452	22.119	(4.900, 81.805)	0.082
Average daily minutes of sedentary activities over the week (mins)	0.961	2.106	(−3.166, 5.088)	0.648
Average daily minutes of lightly active activities over the week ^b^ (mins)	−2.311	1.632	(−5.511, 0.888)	0.157
Average daily minutes of fairly active activity over the week ^b^ (mins)	0.007	0.008	(−0.008, 0.022)	0.345
Average daily minutes of very active activity over the week ^b^ (mins)	0.013	0.007	(−0.001, 0.026)	0.062
Average daily number of stairs climbed over the week ^b^	0.027	0.013	(0.001, 0.052)	0.039

Β = Regression coefficient; SE = Standard error; CI = Confidence level. ^a^ Adjusted for elderly centers, age and percentage of wearing time. ^b^ Logarithm transformation taken.

**Table 4 ijerph-18-10344-t004:** Results from the feedback questionnaire completed by the WAT participants (*n* = 22).

Q	Questions	Every Day		5 to 4 Times a Week				1 to 3 Times a Week				Rarely	
		n	%	n		%		n		%		n	%
Q1	How often do you use the WAT?	20	90.9%	2		9.1%		0		0%		0	0%
Q2	How often do you use the WAT-associated Mobile App (App)?	13	59.1%	1		2.5%		5		12.5%		3	7.5%
Q3	The WAT is easy to use and clear to understand.	Strongly Disagree		Disagree		Somewhat Disagree		Somewhat Agree		Agree		Strongly Agree	
		0	0.0%	2	9.1%	0	0.0%	0	0.0%	16	72.7%	4	18.2%
Q4	The App is easy to use and clear to understand.	0	0.0%	2	9.5%	1	4.8%	2	9.1%	15	68.2%	2	9.1%
Q5	I will continue to use the WAT even after the program.	0	0.0%	1	4.5%	0	0%	2	9.1%	15	68.2%	4	18.2%
Q6	Using worksheets during the face-to-face sessions to formulate a personal exercise plan can enhance my self-efficacy in exercising.	0	0.0%	0	0.0%	1	4.5%	5	22.7%	13	59.1%	3	13.6%
Q7	Using the WAT and the App to formulate personal exercise goals can enhance my self-efficacy in exercising.	0	0.0%	0	0%	0	0%	3	13.6%	16	72.7%	3	13.6%
Q8	Using worksheets during the face-to-face sessions to formulate plans to overcome barriers can enhance my self-efficacy in exercising.	0	0.0%	0	0.0%	1	4.5%	6	27.3%	13	59.1%	2	9.1%
Q9	Setting up personal exercise goals during the face-to-face sessions can enhance my self-efficacy in exercising.	0	0.0%	0	0%	0	0%	6	27.3%	14	63.6%	2	9.1%
Q10	Using the WAT and the App to review my exercise progress (e.g., daily steps, floors climbed, etc.) helps me to maintain habitual physical activity levels.	0	0.0%	0	0%	0	0%	2	9.1%	18	81.8%	2	9.1%
Q11	Using the WAT and the App to revise my exercise plan (e.g., daily steps, floors climbed, etc.) helps me to maintain habitual physical activity levels.	0	0.0%	0	0.0%	0	0%	5	22.7%	14	63.6%	3	13.6%
Q12	The telephone follow-up by the instructor to review my exercise progress helps me to maintain habitual physical activity levels.	0	0.0%	0	0.0%	0	0.0%	1	4.5%	18	81.8%	3	13.6%
Q13	Using the WAT and the App to review the level of achievement of my pre-set exercise goals can enhance my self-efficacy in exercising.	0	0.0%	0	0.0%	0	0%	4	18.2%	14	63.6%	4	18.2%
Q14	The instructor’s positive feedback and suggestions motivate me to engage in regular exercise.	0	0.0%	0	0.0%	0	0%	2	9.1%	16	72.7%	4	18.2%
Q15	Using the weekly exercise record booklet to review my exercise progress helps me to maintain habitual physical activity levels.	0	0.0%	0	0.0%	0	0.0%	3	13.6%	16	72.7%	3	13.6%
Q16	Self-monitoring of my exercise performance from the WAT and the App can enhance my self-efficacy in exercising.	0	0.0%	0	0.0%	0	0.0%	1	4.5%	17	77.3%	4	18.2%
Q17	Using the WAT and the App to observe the changes in my exercise levels helps me to maintain habitual physical activity levels.	0	0.0%	0	0.0%	0	0.0%	3	13.6%	17	77.3%	2	9.1%
Q18	The real-time exercise measurements (e.g., heart rate during exercise) provided by the WAT helps to enhance my self-efficacy in exercising.	0	0.0%	0	0.0%	0	0.0%	3	13.6%	15	68.2%	4	18.2%
Q19	Using the social community functions of the App to share exercise achievements with each other, comment on others’ posts, etc. can enhance my self-efficacy in exercising.	0	0.0%	0	0.0%	1	4.5%	6	27.3%	13	59.1%	2	9.1%
Q20	The positive feedback and encouragement from peers during the face-to-face sessions can enhance my self-efficacy in exercising.	0	0.0%	0	0%	0	0%	4	18.2%	16	72.7%	2	9.1%
Q21	Getting guidance and demonstrations from the instructor can maintain my habitual physical activity levels.	0	0.0%	0	0.0%	0	0.0%	1	4.5%	14	63.6%	7	31.8%
Q22	The “reminder to move” function of the WAT can enhance my self-efficacy in exercising.	0	0.0%	0	0.0%	0	0.0%	6	27.3%	12	54.5%	4	18.2%
Q23	Obtaining appreciation/encouragement for my exercise performance from the support group via the App can enhance my self-efficacy in exercising.	0	0.0%	1	4.5%	0	0.0%	3	13.6%	15	68.2%	3	13.6%
Q24	The badge rewards and celebration vibration reminder from the WAT and the App can enhance my self-efficacy in exercising.	0	0.0%	1	4.5%	0	0.0%	2	9.1%	12	54.5%	7	31.8%
Q25	I would recommend that other older people join this program.	0	0.0%	0	0.0%	0	0.0%	2	9.1%	15	71.4%	4	18.2%

**Table 5 ijerph-18-10344-t005:** Effectiveness of the intervention between groups.

		Mean (SE)				Tests of GEE Model Effects ^a^							Effect Size	Post Hoc Test (*p*)				
Outcome Measures		Baseline (T0)		Post-Intervention (T1)		Time Effect		Group Effect			Group-by-Time Effect			WITHIN-GROUP		Between Groups		
						Wald χ^2^	*p*	Wald χ^2^	*p*		Wald χ^2^	*p*	d	T0 vs. T1		T0		T2
TUG ^b^ (seconds)						10.114	0.006	8.554	0.003		8.879	0.012	0.768			0.027	**	0.002
	Intervention	9.76	(0.53)	7.98	(0.37)									0.006				
	Control	15.62	(1.87)	14.55	(2.27)									0.484				
30-s Chair Stand Test (seconds)						11.566	0.003	4.687	0.030		12.157	0.002	0.687			0.536		0.011
	Intervention	11.36	(0.85)	14.14	(0.81)									<0.001				
	Control	7.50	(1.16)	8.50	(1.49)									0.827				
Two-Minute Walk Test (meters)						11.834	0.003	7.810	0.005		14.898	0.001	1.242			0.156		<0.001
	Intervention	99.84	(4.86)	116.23	(3.30)									<0.001				
	Control	77.11	(6.07)	80.57	(5.71)									0.760				
FFI						6.713	0.035	11.458	0.001		6.562	0.038	0.464			0.048	*	0.007
	Intervention	1.55	(0.14)	1.36	(0.12)									0.234				
	Control	2.44	(0.17)	2.36	(0.20)									0.711				
CSEE						5.422	0.066	8.488	0.004		1.575	0.455	0.284			0.037	*	0.006
	Intervention	43.59	(3.35)	42.32	(2.45)									0.696				
	Control	29.44	(3.70)	23.35	(4.66)									0.090				
BREQ-2 Amotivation						13.380	0.001	0.047	0.828		18.839	<0.001	1.369			0.343		0.345
	Intervention	0.92	(0.40)	1.82	(0.68)									<0.001				
	Control	1.56	(0.68)	1.94	(0.60)									0.310				
BREQ-2 External Regulation						15.614	<0.001	2.930	0.087		3.133	0.209	0.141			0.355		0.232
	Intervention	2.73	(0.64)	4.64	(0.76)									0.106				
	Control	2.28	(0.72)	2.89	(0.92)									0.154				
BREQ-2 Introjected Regulation						2.447	0.294	4.827	0.028		1.268	0.530	0.064			0.062		0.108
	Intervention	3.77	(0.63)	4.64	(0.6)									0.893				
	Control	2.33	(0.72)	3.07	(0.78)									0.807				
BREQ-2 Identified Regulation						6.556	0.038	0.228	0.633		9.564	0.008	0.242			0.385		0.885
	Intervention	11.29	(0.60)	11.39	(0.52)									0.375				
	Control	12.3	(0.44)	10.00	(0.60)									0.034				
BREQ-2 Intrinsic Regulation						4.182	0.124	0.571	0.450		2.819	0.244	0.142			0.593		0.650
	Intervention	10.8	(0.96)	12.18	(0.56)									0.570				
	Control	10.78	(0.92)	11.22	(0.76)									0.108				
MVPA (mins per valid day per week)						0.422	0.810	2.233	0.135		4.310	0.116	0.373			0.292		0.010
	Intervention	218.87	(23.93)	226.53	(24.56)									0.570				
	Control	145.62	(18.25)	126.68	(17.49)									0.041	*			
MVPA (mins per valid day per week, only including > 10 min)						0.192	0.909	3.110	0.078		1.119	0.571	0.156			0.251		0.011
	Intervention	47.76	(9.53)	53.00	(9.66)									0.520				
	Control	24.96	(5.62)	22.36	(5.38)									0.421				
Maximum duration of brisk walking ^#^ over the week (mins)						8.256	0.016	4.278	0.039	*	2.904	0.234	0.559			0.158		0.025
	Intervention	9.23	(1.88)	9.00	(2.71)									0.925				
	Control	5.78	(1.51)	3.17	(1.58)									0.073				
Average duration of brisk walking ^#^ per week ^b^ (mins)						6.720	0.035	5.074	0.024	*	1.346	0.510	0.121			0.091		0.019
	Intervention	3.31	(0.83)	3.75	(1.23)									0.687				
	Control	1.84	(0.65)	1.30	(0.78)									0.421				
Peak cadence per week (steps/min)						7.270	0.026	2.137	0.144		1.409	0.494	0.248			0.579		0.072
	Intervention	116.27	(4.14)	112.05	(3.73)									0.428				
	Control	110.83	(3.92)	104.42	(4.97)									0.014				
Average step count per week (steps)						0.611	0.737	2.210	0.137		2.043	0.360	0.337			0.605		0.040
	Intervention	12,787.32	(908.64)	13,683.30	(845.53)									0.221				
	Control	10,352.90	(752.45)	10,264.35	(642.43)									0.494				

* *p* < 0.05, ** *p* < 0.01. SE, standard error; TUG, Timed Up and Go Test; FFI, Fried frailty index; CSEE, Chinese self-efficacy for exercise; C-BREQ-2, Chinese Behavioral Regulation in Exercise Questionnaire-2; MVPA, Moderate to vigorous physical activity. ^#^ Brisk walking was defined as >100 step/min. ^a^ Adjusted for elderly centers, age, sex, and walking aids. ^b^ Logarithm transformation taken.

**Table 6 ijerph-18-10344-t006:** Effectiveness of the intervention for the intervention group over time analyzed by GEE ^a^.

Outcome Measures [Mean (SE)]	Baseline (T0)		Post-Intervention (T1)		Follow-Up				Tests of Time Effects		Effect Size (d)	Post Hoc Test (*p*)					
					(T2)		(T3)		Wald χ^2^	*p*		T0 vs. T1	T0 vs. T2	T0 vs. T3	T1 vs. T2	T1 vs. T3	T2 vs. T3
TUG ^b^	9.76	(0.53)	7.98	(0.37)	7.62	(0.30)	8.23	(0.45)	30.538	<0.001	0.673	<0.001	<0.001	0.003	0.128	0.283	0.014
30-s Chair Stand Test	11.36	(0.85)	14.14	(0.81)	13.40	(0.83)	15.22	(1.35)	29.995	<0.001	0.710	<0.001	0.001	0.001	0.431	0.267	0.060
Two-Minute Walk Test	99.84	(4.86)	116.23	(3.30)	115.73	(5.14)	112.00	(5.87)	24.868	<0.001	0.459	<0.001	0.001	0.039	0.828	0.302	0.071
FFI	1.55	(0.14)	1.36	(0.12)	1.20	(0.14)	1.22	(0.15)	18.579	0.001	0.557	0.234	0.055	0.018	0.344	0.289	0.996
CSEE	43.59	(3.35)	42.32	(2.45)	46.15	(3.54)	46.89	(3.19)	5.038	0.283	0.228	0.696	0.353	0.392	0.234	0.177	0.986
*BREQ-2*																	
Amotivation	0.92	(0.40)	1.82	(0.68)	3.50	(0.84)	3.05	(0.60)	25.395	<0.001	0.807	<0.001	0.002	<0.001	0.512	0.203	0.622
External Regulation	2.73	(0.64)	4.64	(0.76)	4.50	(0.96)	4.75	(0.68)	10.395	0.034	0.527	0.106	0.015	0.023	0.752	0.882	0.527
Introjected Regulation	3.77	(0.63)	4.64	(0.60)	3.86	(0.96)	5.25	(0.45)	11.423	0.022	0.402	0.893	0.133	0.063	0.154	0.038 *	0.517
Identified Regulation	11.29	(0.60)	11.39	(0.52)	10.64	(0.44)	11.75	(0.52)	10.810	0.029	0.111	0.375	0.495	0.773	0.002 **	0.100	0.665
Intrinsic Regulation	10.82	(0.96)	12.18	(0.56)	11.36	(0.44)	12.80	(0.52)	9.057	0.060	0.284	0.570	0.049	0.268	0.023 *	0.394	0.053
MVPA (mins per valid day per week)	218.87	(23.93)	226.53	(24.56)	206.03	(22.78)	197.52	(27.02)	5.189	0.268	0.469	0.605	0.339	0.057	0.228	0.088	0.240
MVPA (mins per valid day per week, only including >10 min	47.76	(9.53)	53.00	(9.66)	44.43	(10.22)	47.97	(10.90)	1.341	0.854	0.199	0.541	0.772	0.960	0.396	0.519	0.742
Maximum duration of brisk walking over the week ^b^	9.23	(1.88)	9.00	(2.71)	10.89	(3.08)	10.89	(2.49)	16.295	0.003	0.281	0.935	0.178	0.236	0.146	0.190	0.621
Average duration of brisk walking per week ^b^	3.31	(0.83)	3.75	(1.23)	3.51	(0.99)	4.47	(1.07)	8.039	0.090	0.297	0.717	0.316	0.113	0.819	0.386	0.337
Peak cadence per week (steps/min)	116.27	(4.14)	112.05	(3.73)	108.26	(4.23)	114.11	(3.35)	9.567	0.048	0.060	0.186	0.020	0.811	0.478	0.338	0.035
Average step count per week	12,787.32	(908.64)	13,683.30	(845.53)	12,746.26	(825.69)	12,380.51	(1136.08)	2.995	0.559	0.281	0.206	0.942	0.489	0.182	0.111	0.381

* *p* < 0.05, ** *p* < 0.01. TUG, Timed Up and Go Test; FFI, Fried frailty index; CSEE, Chinese self-efficacy for exercise; C-BREQ-2, Chinese Behavioral Regulation in Exercise Questionnaire-2; MVPA, Moderate to vigorous physical activity. SE = standard error. ^a^ Adjusted for elderly centers, age, and sex. ^b^ Logarithm transformation taken.

## Data Availability

Data is not available for public access.

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
