# Peer review of "Enhancing the Physical Activity Levels of Frail Older Adults with a Wearable Activity Tracker-Based Exercise Intervention: A Pilot Cluster Randomized Controlled Trial"

_ijerph, 2021, doi:10.3390/ijerph181910344_

Round 1

Reviewer 1 Report

The manuscript deals with the feasibility and the effects of a wearable activity tracker -based exercise intervention in frail older individuals. This is an interesting and important study. However, I have some comments, which I would like to be addressed.

Major point:

The sample participated in the study is heterogeneous, which may affect the results. Baseline characteristics shouldn’t have any statistically significant difference, so that the results would have been comparable. On the contrary, there is a statistically significant difference in the age, sex and walking aids between the two groups. The younger males with no walking aids participated in the intervention group, while the opposite in the control group. There seem to be an eight-year difference, which is a lot at that age. Thus, there is a question whether even the same intervention has the same effect in a group of 72 and a group of 80 years individuals. These two groups are not comparable, and the second group cannot serve as a control group of the intervention group. This is an issue that cannot change. At least, this should be pointed out in the limitation part.

Minor points:

  1. The text is too long.
    1. Authors should be strict with the text structure. For example: The part WAT-based components in the intervention group included not only the method used, but also, literature review, results, and discussion.
    2. Authors should avoid repetition. e.g. lines 183-185 are similar to 51-55. Moreover, the part “implementation procedures” are described in the previous parts, in pages 153-158 and in figure 1. Since physical training in the control group was the same (lines 191-204) this part can be omitted. Differences can be incorporated into the previous part. Randomization (2.6) is also repeated on page 3.
  2. The timeline is described differently in several parts and is confusing. Please clarify the lines 226 and 227. It is mentioned that measurements at baseline are called (T0), one week is (T1), and after the completion of the intervention is what? Moreover, in lines 315 and 316 there are also T2 and T3 mentioned. Please include all timepoints in a section called research design. In this section, randomization can also be included.
  3. Why is 2.4.3. called “Preliminary effect outcomes”? These are not outcomes, since no intervention is done. I suppose authors mean baseline measurements, but still these measurements were repeated at different timepoints
  4. Line 109-110 is the same with 111-112
  5. In Figure 1, I suppose FTF is (face to face), but please explain all abbreviations at the end of the figure. Please provide the units of all measurements in the tables
  6. Please, name the abbreviations before using them. What dose RA mean in line 253
  7. Lines 512, 529, 533 and 579, please change “all of the …” into “all’’, e.g. all participants…

Author Response

Dear Reviewer,

We are very grateful for your valuable comments on how to improve our manuscript. Our response to each comment is listed in the attached file. We look forward for your kind consideration of the publication of this manuscript.

Regards,

Authors

Reviewer 2 Report

IJERPH-1374308 presents results of a pilot randomized controlled trial for enhancing PA in frail older adults. While some parts of this paper were interesting, other areas could be improved. I hope the authors consider my feedback.

MAJOR COMMENTS

  • Introduction: In lines 61-72, it should be clearer how the WATs listed differ from traditional accelerometers such as ActiGraph. Otherwise, WAT in this regard are not overly novel given that accelerometers are very common in PA measurement for free-living settings.
  • Lines 73-101: This whole paragraph could be revised for concision. Just be more to the point about driving the purposes and consider moving away from hypothesis statements.
  • Lines 122-124: Please justify why these exclusions were made in the text because this overlaps with how frailty may have been defined.
  • Section 2.4: Please justify in the text why these time points were selected for outcome effect.
  • Limitations: There was also no clear control group.

MINOR COMMENTS

  • Title: No need to include “(WAT)” in the title.
  • Maybe instead consider stating, “frail older adults” instead of “frail older people”. This language is generally more accepted in the literature.
  • Lines 44-46: This PA recommendation might be a little dated. Please revise this sentence with a more updated recommendation and citation.
  • Figure 1: Can this figure be revised a little? Specifically, how “week” is presented with the other figure components. Physical training for example does not fit into weeks 1 or 2. Please also make sure all figures and tables stand alone by defining any abbreviations.
  • Line 153: Avoid re-presenting figures after they are first introduced.
  • Considering re-structuring some parts of the Methods for prospective flow. For example, Sections 2.5-2.7 could have come a bit earlier in this section.
  • Methods: Make IRB approval and written informed consent is included in this section, not just in the note at the bottom of the paper.
  • Section 3.2: No need to include statistics such as X2 because they already inform p-values. Instead, maybe consider presenting the actual differences alongside p-values. This comment is generalizable to other parts of the results section.
  • Tables: Maybe instead flip the “total” column with “intervention”. This will make comparisons clearer.
  • Table 1: P-values can’t = 0.000. Please revise as <0.001 where appropriate throughout the paper.
  • Make any changes to the abstract that align with those made to the text.

Author Response

(The authors gave the same response as above.)

Round 2

Reviewer 1 Report

I am pleased with the revised version. 

Author Response

Thanks for the approval for publication.

Reviewer 2 Report

The authors have done a nice job addressing my previous concerns, but the paper would benefit from some additional revisions.

***Table 1: Having both p-values and "*" are repetitive. Just list the p-values given those are more precise. The "*" can be deleted.

***Table 3: Wald chi-squared and "*" can be deleted.  

Author Response

Thanks for the further comments, we have revised accordingly. 
